# Pet dogs’ Behavioural Reaction to Their Caregiver’s Interactions with a Third Party: Join in or Interrupt?

**DOI:** 10.3390/ani12121574

**Published:** 2022-06-18

**Authors:** Sabrina Karl, Kristina Anderle, Christoph J. Völter, Zsófia Virányi

**Affiliations:** Comparative Cognition, Messerli Research Institute, University of Veterinary Medicine Vienna, Medical University of Vienna, University of Vienna, 1210 Vienna, Austria; kristina.anderle@gmx.at (K.A.); christoph.voelter@vetmeduni.ac.at (C.J.V.); zsofia.viranyi@vetmeduni.ac.at (Z.V.)

**Keywords:** domestic dog, attachment, emotions, jealousy, behavioural synchronization, human–fake dog interaction

## Abstract

**Simple Summary:**

Behavioural studies in dogs have shown that dogs form strong bonds with their caregivers and display attachment behaviours towards them, especially in stressful situations. Some observational studies in dogs and dog caregivers frequently reported the occurrence of jealousy-like behaviours, particularly during affiliative caregivers’ interactions with other dogs; so-called social rivals. Thus far, due to contradictory results, this phenomenon remains unclear. In this study, we investigated pet dogs’ behavioural reactions during two different types of interactions (greeting vs. examining) performed either by the dog’s caregiver or a stranger with a remotely controlled, realistic-looking fake dog. We predicted that the dogs would rather respond in a jealousy-consistent manner when observing their caregiver interacting in an affiliative way with the fake dog. During the tests, the dogs observed the interaction at first and could later join in. We found that the dogs’ initial neutral or negative reaction towards the fake dog changed to a more positive reaction during the affiliative interaction of the caregiver with the fake dog. When joining in with the interaction, more dogs displayed friendly behaviours towards the fake dog when the caregiver was present, but they also tried to block the interaction more often, as compared to the stranger–fake dog interaction. Taken together, we did not observe clear jealousy-like behaviours in the dogs but found indicators of social referencing and behavioural synchronization of the dogs with their caregivers which supports the assumption that human affiliative behaviours towards others can facilitate dogs’ positive reactions.

**Abstract:**

Pet dogs are promising candidates to study attachment-related and potentially jealousy-like behaviours in non-human animals, as they form a strong and stable bond with their human caregivers who often engage in affiliative interactions with diverse social partners. Nevertheless, it is still debated whether non-human animals are capable of experiencing such complex emotions. Even though caregivers frequently report observations of jealousy-like behaviours in dogs, behavioural studies in dogs have thus far led to contradictory results. Adding to this complexity, dogs appear extraordinarily skilled in understanding humans’ communicative behaviour and can flexibly and diversely interact with them in social contexts. Here, we aimed at investigating (1) whether dogs indeed respond in a jealousy-consistent manner when seeing their caregiver interact in an affiliative way with a remotely controlled, realistic-looking fake dog, or (2) whether they would rather synchronize their reaction to the fake dog with the caregiver’s behaviour, or (3) whether they respond directly to the caregiver without paying much attention to the third party. To address what drives the dogs’ behaviours in this triadic situation, we compared four groups of dogs who first observed and then joined the interaction of either the caregiver or a stranger greeting or medically examining the fake dog. We found that the dogs initially responded negatively or neutrally when the fake dog entered the room but changed to more positive reactions when the caregiver approached the fake dog, especially if initiating a positive interaction. When being released, more dogs showed friendly behaviours towards the fake dog when the caregiver—rather than the stranger—was interacting with it. At the same time, however, the dogs tried to block the interaction of the caregiver with the fake dog more often than the one of the stranger. In conclusion, we did not find clear evidence for jealousy-like behaviours in dogs during the human–fake dog interactions, but we observed indicators of behavioural synchronization with the caregivers, suggesting that the caregivers’ affiliative behaviours directed at a third party may more often facilitate positive than negative interactions in dogs.

## 1. Introduction

Whereas the study of human emotions has been greatly facilitated by using self-reports, this method is obviously not available to investigate whether and how non-human animals experience basic and complex emotions [1,2]. Accordingly, one way to assess the occurrence of emotion-driven behaviours in companion animals is to conduct survey studies interviewing their caregivers.

This method has been frequently used with regard to emotion-driven behaviours in pet dogs, and often found that the caregivers typically reported jealousy as one of the most prevalent emotions driving dogs’ social interactions besides the six basic emotions (joy, anger, fear, disgust, sadness, surprise) and complex or social emotions such as disappointment, shame and compassion [1,3,4]. As an example, one study [1] surveyed 1.023 Dutch-speaking dog and/or cat caregivers, and found that, of all reported emotions, joy, fear and jealousy were the emotions most frequently attributed to dogs and cats. Similarly, a survey [2] reported very high scores of dog caregivers (N = 337) having observed jealousy-like behaviours, i.e., in 81% of the dogs. In a second survey where they conducted in-depth structured interviews with 40 dog caregivers, they further investigated the displayed behaviours, contexts and consistency of jealousy, and reported that all caregivers stated that their dogs could get jealous, especially when the caregiver was interacting with another person (50% of participants) or another dog or animal (45%). In all cases, the consistent reason for jealousy-like dog behaviour was the caregiver’s attention towards the third party, particularly when affection was also expressed (25% of reports), e.g., cuddling (22%). The most frequently reported jealous behaviour in dogs was attention seeking, in particular by pushing against the caregiver (in 50% of reports), whining, growling or barking (40%), and occasionally aggressive behaviours, e.g., biting the other dog. The contexts and behaviours reported in that study were consistent with the ones stated in common definitions of jealousy in the literature. More specifically, jealousy is mostly described as a complex, secondary emotion that facilitates the maintenance of a valuable, social relationship. It especially emerges when this special bond is threatened by a third-party, a so-called social rival, and is often accompanied by feelings of discomfort and distress in the affected observer and/or intervening behaviours [2,4,5,6].

In line with the above mentioned reports, it has been suggested that pet dogs may be a promising model species to investigate jealousy in non-human animals [6,7,8,9]. Given that pet dogs closely cohabit with humans, form strong and stable attachment bonds with them which highly resemble the human mother–child bond [10,11,12,13,14,15,16,17,18], and typically have plenty of opportunities to observe their caregiver engaging in positive social interactions with others, there are many occasions to witness them in situations that typically evoke jealous reactions in human children [19,20,21].

Despite their presumed functional relevance and in contrast to the above-mentioned findings of surveys, behavioural studies using objective methods to assess jealousy-like behaviours in pet dogs have brought rather contradictory results thus far [6,8,22,23]. Some authors reported distinctive jealousy-like behaviours such as blocking the social rival (fake or real dog) from approaching the caregiver or getting (sometimes aggressively) between them during their interaction, presumably to intervene or try to separate their valuable caregiver from the intruder [6,8]. Using a different approach, the most recent study presented human–fake dog interactions to leashed pet dogs while measuring their pull force towards the scene [7]. Leash pulling was measurably stronger when the caregiver engaged in such an interaction, which was interpreted as evidence for jealousy.

Other studies, however, reported a lack of evidence for jealousy-like behaviours, e.g., aggression, in dogs that observed either the caregiver or a stranger interacting with non-social objects and fake or real dogs [22,23]. One of these studies was conducted at home with caregivers of two dogs living in the same household, assessing the dogs’ behavioural patterns while the caregiver ignored one of the dogs and expressed affection towards the other one. The tested dogs showed generally low stress levels, suggesting that the test situation was not perceived as very stressful, and some dogs displayed friendly behaviours (e.g., sniffing, licking) as well as more rough behaviours (e.g., pushing or biting) towards the companion dog in both the test and control (i.e., both dogs were ignored) conditions [22]. In another study using fMRI in awake dogs, the authors found increased amygdala activation while the dogs were observing their caregivers feeding a fake dog [9]. Importantly, especially dogs with a high dog–dog aggression score (Canine Behavioral Assessment and Research Questionnaire—C-BARQ; [24]) showed this particular amygdala activation. It remains unclear whether this specific finding was based on potential jealous emotions in the tested dogs or due to the involved food delivery (e.g., evoking resource defence) or the fake dog itself (e.g., grounded on dog–dog aggression, namely, threat or fear regarding other dogs) which could result in similar behavioural responses, i.e., fear- or defence-induced aggression.

The mismatch between behavioural studies and caregiver reports with regard to the prevalence of jealousy-like behaviours in dogs could be due to a general bias of caregivers to anthropomorphise their dog’s behaviour. Other reasons for the “jealousy bias” in caregiver reports range from a general negativity bias of humans who tend to give a larger weight to behaviours motivated by negative rather than by positive emotions [25,26] to the caregivers’ special sensitivity to their dogs’ emotions [10,12,14,18,27,28]. Finally, an additional practical reason for the same bias might be that caregivers watch out more carefully for these behaviours, as they may signal potential conflicts where the caregivers eventually need to intervene.

Given the possibility that human observers may be biased to interpret dog behaviour in terms of jealousy during triadic social interactions, in the current study we aimed to investigate the behavioural responses of pet dogs using a more diverse approach and a situation that in principle could evoke jealousy-like behaviours, but where the dogs’ behaviours may also be driven by other motivations, such as trying to engage with an obviously interactive human partner or to synchronize with her actions. To this end, we compared the behaviour of four groups of dogs that could observe their caregiver or a stranger (unfamiliar person) while they interacted with a realistic-looking fake dog in a positive or neutral manner (i.e., engaging in greeting vs. a simple veterinary examination, henceforth referred to as “vet check”). Following a short observation period, the subject dog was released to freely join the ongoing interaction.

Given these four experimental situations, we predicted that if dogs respond with jealousy or jealousy-like emotions to seeing a positive caregiver–fake dog interaction, they would show more negative, offensive reactions towards the fake dog such as blocking its interactions with the human partner. Given that these reactions should aim at protecting and maintaining a valuable relationship, we expected that the dogs would display these behaviours more around the caregiver than around the stranger and more often during greeting than during a vet check.

In contrast, if dogs would want to synchronize with their human partners, their interactions would still be directed at the fake dog but would not aim at blocking its interactions with the human. Behavioural synchronization is defined as the temporal matching of actions, movements or gestures of several individuals, in particular in social species and cooperative breeders, and is, for example, related to increased social cohesion and attachment [29,30,31,32,33]. Interestingly, this phenomenon can be observed within and between species [34,35,36,37] and it facilitates achieving a common goal [38] or increasing affiliation to strengthen relationships [39], e.g., while walking next to each other in daily life [40,41,42]. Behavioural studies showed that dogs actively adapt to and synchronize their behaviour with a human partner, and do so more with their human caregivers than with strangers, indicating the role of affiliative relationships in this process [43,44]. Social referencing is one form of such behavioural synchronization; here dogs seek emotional information from their caregivers and mirror their behaviour when being confronted with a novel object. More specifically, in two social referencing studies, the tested dogs approached a novel object (i.e., a fan) faster, stayed closer to it after having received a positive cue about it from their caregiver and then mirrored the caregiver’s behaviour; when observing a negative emotional response of the caregiver, they approached it slower and moved away from the fan like the caregiver [45,46]. When a stranger was present in the same situation, the dogs performed referential looking at the stranger to gain information as well, but less than they did with the caregiver (60% vs. 76%). Given that dogs show this behavioural synchronization more with their caregiver than with strangers, we expected that dogs would show a positive, friendly reaction towards the fake dog longer in the greeting than in the vet check condition, and that this differentiation would be stronger when the caregiver is interacting with the fake dog as compared to the stranger.

Finally, it is also possible that dogs simply respond to the behaviour of humans who show readiness to interact with another dog, and approach them directly without engaging much with the third party. For example, one study [47] demonstrated that dogs readily distinguish between two humans and approach them differently if they have shown generous or selfish reactions to a third party asking them for food. Dogs are well-known for responding flexibly to various human behaviours, including tolerant and friendly reactions to a person approaching them in a friendly manner and being rather avoidant and aggressive in response to a threatening approach [48]. Moreover, researchers [49] demonstrated that dogs’ behavioural responses to such a threatening approach also depended on the context, as they showed tolerant, contact-seeking responses towards a threatening stranger if a playful context had been set up in advance, whereas in a non-playful situation they remained avoidant and aggressive. This differential response was, however, less apparent when the caregiver approached the dogs in a threatening manner, as the dogs seemed to have interpreted this as a form of play and showed a contact-seeking response, independent of the context set up in advance. As these findings suggest a more differential response to a stranger’s behaviour, we expected that the dogs would react more positively to the greeting than to the vet check condition, especially when the stranger was offering these interactions to the fake dog.

Given that these three hypotheses predict different behaviours and also various differentiation across our four test situations (see Table 4 for an overview of predictions), by recording the dogs’ behaviours towards the fake dog and the human partner we aimed at investigating whether the dogs’ reactions corresponded to our predictions: (1) if they experienced jealousy-like emotions and performed behaviours that, in humans, would be consistent with this feeling, e.g., blocking or intruding, (2) if they tried to synchronize their behaviour with that of the caregiver or (3) if they simply directly responded to the humans’ behaviour.

In addition to the behavioural analysis, we asked the caregivers to fill in a short questionnaire about their daily observations of potential jealousy-like behaviours in their dogs, to investigate whether the reported prevalence of jealousy in our sample was consistent with former studies.

## 2. Materials and Methods

### 2.1. Subjects

All our tested dogs (N = 150) were privately owned pet dogs and were recruited from human caregivers who live in Vienna and nearby via the Clever Dog Lab website and database or via social media announcements. Forty-eight of the tested dogs had to be excluded from the data analyses because of initial setup changes and technical or execution errors during the test procedure. Thus, the final sample consisted of 102 adult pet dogs of both sexes (51 males, 51 females), different ages (mean age: 5.8, range: 10 months to 12 years), and various breeds (see Appendix A for details). In the caregiver greeting group (CG) we tested 25 dogs, in the caregiver vet check group (CV) 26, in the stranger greeting group (SG) 26 and in the stranger vet check group (SV) 25 dogs (see Table 1). The dogs were allocated to the groups in a pseudo-random manner while counterbalancing the four test groups with respect to age and sex.

### 2.2. Experimental Setup

All experimental tests were conducted in a test room (7.25 m × 6.05 m) at the Clever Dog Lab, Vienna, and were recorded by four video cameras (two hand-held cameras, JVC GZ-MG750; two wide-angle H.R. Colour CLD cameras) installed in the room corners (height: 2.00 m). The video cameras were connected to a desk computer and a monitor placed outside the test room. The test room had two doors along one side of the room and two windows at the opposite side. Between the two windows, a hook was fixed to the wall at an equal distance to both doors, and a chair was located either on the left or right side of the hook for either the caregiver or the experimenter (E1) sitting next to the dog during the test procedure. The side of the chair was counterbalanced across the tested dogs depending on the door through which the fake dog entered. The person sitting on the chair next to the dog wore sunglasses throughout the Introduction, Interaction and Reaction phases. Pieces of insulating tape placed on the floor marked five spots to make sure that all test participants were located in the correct positions during the entire test procedure. Additionally, red tape marks were placed on each door and window, and the humans were instructed to look at these during the test trial.

To investigate how the subjects reacted to a human’s interaction with another dog, we used a fake dog (Melissa & Doug Labrador; length 90 cm, withers height 35 cm) that was mounted on a board with wheels (see Figure 1). The fake dog was remotely operated by a second experimenter (E2) watching the test procedure via the computer monitor placed outside of the room. This person was also equipped with a timer to measure the exact time of each test phase and headphones to tell the start and end of the respective test phase to E1 inside the room. E1 was either interacting with the fake dog as the stranger or sitting on the chair next to the subject dog. Taken together, both the caregiver and E1 (stranger) were always in the test room whereas E2 was standing outside, operating the fake dog and measuring the test phase times while observing the entire test trial. Specifically, two female persons acted as the two experimenters throughout the study, and their roles were counterbalanced and pseudo-randomized across dogs within each group. During the entire test procedure, the caregivers were in the test room and could observe their dogs. The caregivers were allowed to terminate the experiment at any point of the test trial if they had the feeling it might be too stressful for their dog. This happened only a few times (N = 5), and the test trials were immediately stopped. All subject dogs wore harnesses during the test procedure to avoid any pressure on their neck while being leashed. They were provided with fresh water before and after the test.

### 2.3. Experimental Design and Test Groups

For the experimental design, we used a 2 × 2 factorial, between-subject design conducted with two humans (caregiver vs. stranger) interacting in an either positive (i.e., greeting and cuddling) or neutral (i.e., veterinary check, examining ears and teeth—called vet check) manner (see Figure 2). During our tests each subject dog was tested in a single trial in one of four different groups: (a) caregiver greeting (CG), (b) caregiver vet check (CV), (c) stranger greeting (SG) and (d) stranger vet check (SV).

In the greeting groups (i.e., CG, SG), either the dog’s caregiver or E1 greeted and petted the fake dog. To make sure that the stranger’s greeting interaction style with the fake dog was similar to the caregiver’s style, the caregiver was asked to demonstrate how they usually greet their dog before the actual test. Having watched the individual caregiver’s greeting style, E1 tried to copy this style as similar as possible during the test with the fake dog. In the vet check groups (i.e., CV, SV), either the dog’s caregiver or E1 examined the fake dog’s ears and teeth by first lifting and inspecting the ears and then pretending to raise its lips and check its teeth. During all interactions the humans had been asked to talk continuously to the fake dog and remained in the same location and body posture (kneeling or sitting on the floor frontal to the fake dog; see Figure 2). During the greeting condition the caregiver or E1 were talking in a friendly, cheerful manner to the fake dog (with a positive facial expression), whereas in the vet check condition the talking was calmer and less enthusiastic (with a neutral facial expression) in comparison.

### 2.4. Test Procedure

Before the actual test, the dog’s caregiver received detailed, verbal instructions about how to perform during the entire test trial and they watched an exemplary video of the respective human–fake dog interaction (i.e., greeting or vet check). During this first introduction, performed by the same experimenter in all tests, the experimenter never interacted with the subject dog. Before conducting the test, the fake dog was located outside the test room, out of the subject dog’s view. The two experimenters communicated via headphones and their mobile phones during the test trial. Each trial started when the experimenter and the caregiver with their leashed dog entering the test room for the Exploration phase to habituate the dog to the test setup (duration: 4 min). After this, the Introduction (duration: 7 s), Interaction (duration: 10 s) and Reaction phases (duration: 3 min) of the test followed (see Figure 3).

**Exploration phase:** The Exploration phase lasted four minutes and served to familiarize the dogs with the test room and setup. E1, the caregiver and the dog entered the room and when the door was closed the dog was released to move freely through the test room. Then, E1 and the caregiver walked to predetermined spots in the middle of the room and calmly talked to each other, without gesturing and while ignoring the dog. After four minutes, E2 informed E1 about the end of the phase, at which point E1 asked the caregiver to call the dog and leash it to the hook at the wall. After this, the caregiver either left the room and E1 sat down on the chair next to the leashed dog in the caregiver groups (CG, CV), or the caregiver sat down on the chair next to the dog and E1 left the room (in the stranger groups—SP, SV). The person that sat on the chair immediately put on sunglasses and looked straight at a mark on the opposite door to avoid looking at or interacting with the dog.

**Introduction phase:** Depending on the test group, either E1 (SP, SV) or the caregiver (CG, CV) slowly entered the test room simultaneously with the fake dog that was remotely operated by E2 staying outside. After entering, E1 closed her door and E2 closed the door behind the fake dog, and the human and the fake dog resumed moving straight to a predetermined spot on the floor. Here they stopped and remained standing still frontally towards the subject dog. The human looked at a mark on the opposite window to avoid looking at the subject dog. The Introduction phase started when the human and the fake dog stopped at this position; it lasted for seven seconds.

**Interaction phase:** After the Introduction phase, the human (caregiver or experimenter as stranger, depending on test group) and the fake dog turned and moved simultaneously towards each other until they reached a predetermined spot in the middle. Then, the human kneeled down and started the interaction with the fake dog in accordance with the respective test group. After the person interacted with the fake dog for 10 s, E2 informed E1 via headphone that the phase ended. At this point, E1 released the dog from the leash in the caregiver groups, or she gave a head signal (nodding) to the caregiver to unleash the dog in the stranger groups. The person who unleashed the dog kept on sitting in the chair and said “OK” to the dog to let it know that it can move freely.

**Reaction phase:** Whereas during the Interaction phase the subject dog was leashed and could only observe the human–fake dog interaction, in the Reaction phase the dog was released and free to join the interaction. After the Interaction phase, the Reaction phase directly followed, and the human–fake dog interaction was continuously pursued during both phases. The person interacting with the fake dog did not pay any attention to the approaching subject dog. The end of the phase was again communicated by E2 via headphones to E1 who then either stopped the interaction with the fake dog (SP, SV) or stood up from the chair (CG, CV) and ended the test. Then, the caregiver called and leashed the dog and left the test room. The Reaction phase lasted three minutes.

### 2.5. Behavioural Coding

All tests were recorded on video, and the dogs’ behaviours and behavioural categories were manually scored using the coding tool of Loopy (http://loopb.io, Loopbio Gmbh, 1080 Vienna, AUSTRIA) later on (see Table 2). To describe the overall behavioural reaction of each dog, they were categorised in each phase as showing either a friendly, neutral, insecure, insecure-offensive or offensive reaction to the fake dog (see Table 3). If the dogs had shown an either friendly or offensive reaction during this approach at any point, they were categorised as such. Importantly, in the Reaction phase, we categorized the dogs based on the behaviour they showed during their first approach towards the fake dog (except for 13 dogs that never approached the fake dog at all). Additionally, we coded whether the dogs approached the fake dog or the human member of the dyad first, whether or not they showed friendly behaviours towards the human (caregiver/stranger) interacting with the fake dog and towards the fake dog itself, and whether they expressed blocking behaviour. Furthermore, we coded how often the dogs sniffed the anal region of the fake dog.

To ensure reliability of the video coding, 50% of the test videos (N = 51) were coded by a second coder.

### 2.6. Statistical Analysis

For the behavioural categories in the Introduction and Interaction phases, we modelled the frequencies of these categories by fitting a mixed ordinal regression model [50] using function ‘clmm’ of the R package ordinal [51]. We included the predictor variables of phase (Introduction/Interaction), human (caregiver/stranger), treatment (vet check/greeting), age (in months, z-transformed) and sex. We also included all possible interactions between phase, human and treatment. In case of non-significant interactions, we dropped the interactions from the models in order to evaluate the main effects. Additionally, we included subject ID as a random effect and the random slope of the phase within subject ID. For the Reaction phase analysis, we fitted an ordinal regression model (using function clm) with the same predictor variables except for phase.

We analysed the behavioural data in the Reaction phase using generalised linear models (GLM) in R [52], using R packages lme4 [53] and MASS [54]. The response variables of first approach to fake dog, interaction with fake dog, interaction with human and blocking were binary (present/absent) and we fitted models with a binomial error structure. The anal sniffing response was analysed as a count variable (number of observed anal sniffing instances per dog) using a negative binomial GLM. In all of these models, we included the test predictor variables of human (caregiver/stranger), treatment (vet check/greeting) and their interaction, as well as age (in months, z-transformed) and sex as control predictors. In case of non-significant interactions, we dropped the interactions from the models in order to evaluate the main effects. For all models, we first established that the model was significant when compared to the null model only when including the control predictors of age and sex [55]. We used likelihood ratio tests to conduct the full-null model comparison and to test the effects of individual predictor variables [56].

We assessed the model stability by excluding one subject at a time, refitting the models and comparing these models to the original model. This procedure revealed the models to be stable with regard to the fixed effects. Moreover, we calculated variance inflation factors (VIF) [57] to check for collinearity among the predictor variables, which revealed that there were no collinearity issues (maximum VIF < 1.1). For the count response variable, we checked for overdispersion. A Poisson model was overdispersed (overdispersion parameter: 1.98); therefore, we fitted a negative-binomial model instead (overdispersion parameter: 1.28).

### 2.7. Caregiver Jealousy Questionnaire

To assess whether the dogs’ caregivers observed or experienced jealousy-like behaviours in their dogs towards other dogs, we asked the caregivers to fill in an online questionnaire consisting of four jealousy-related questions before the actual test was conducted (see Appendix A). In detail, we asked the caregivers whether their dogs reacted jealousy-like towards other dogs, and if so, which behaviours the dogs showed in these situations, and when and where they reacted jealousy-like towards conspecifics.

## 3. Results

### 3.1. Caregiver Jealousy Questionnaire

Based on the questionnaires that the caregivers filled in about their everyday observations of jealousy-like behaviours in their dogs (see Appendix A), we found that 71 (70%) of the caregivers (N = 102) reported that their dogs usually reacted jealousy-like towards other dogs. They reported that their dogs usually displayed jealousy-like behaviours such as squeezing in-between the human–dog interaction and trying to separate the caregiver from the other dog, reacting aggressively by growling, snarling, barking, launching and snapping at the other dog, trying to regain the caregiver’s attention and affection, e.g., by jumping at or licking him/her, showing displacement behaviours such as play bowing and sniffing or stress induced signals such as lip licking, whining, yawning, screaming or displaying piloerection and tail-wagging. Thirty-one (30%) caregivers stated that their dogs were not jealous towards conspecifics but eight of them indicated that their dogs sometimes showed jealousy-like behaviours such as behaving “nervous” and “insecure”, trying to regain the caregiver’s attention and affection by squeezing in-between the interaction, jumping at the caregiver, launching at the other dog or chasing it away, or displaying “sad”/“insulted” expressions.

Seventy-one of the (jealousy experiencing) dog caregivers stated that the dogs’ main reasons for jealousy-like behaviours were greeting, praising, training, playing with or talking to another dog, or when the other dog approached or was too close to the caregiver. Regarding the question where jealousy-like behaviours mainly occurred, 37 (52%) caregivers said at home, 16 (23%) during dog walks, 16 (23%) in dog zones and 7 (10%) stated that these behaviours occurred location-independently. Seven (10%) caregivers reported that it differed a lot and could also be dog or human dependent. Two (3%) stated it mainly happened in the other dog’s home or one caregiver specified the dog school as the main location (see Appendix A for details). Note that the caregivers could give multiple answers to this question.

### 3.2. Behaviours Observed and Inter-Observer Reliability

The inter-observer reliability (IOR) was assessed by using Cohen’s kappa and Spearman’s correlations: behavioural categories (Cohen’s kappa = 0.79), fake dog approach (Cohen’s kappa = 0.79), friendly interaction with the fake dog (Cohen’s kappa = 0.52) and the human (Cohen’s kappa = 0.80), blocking behaviour (Cohen’s kappa = 0.86) and sniffing of the fake dog’s anal region (Spearman = 0.83). The IOR of non-offensive fake dog manipulations and agonistic behaviour such as dominant behaviours were too low (<0.50) and, therefore, were excluded from the data analysis. Offensive manipulation behaviours towards the fake dog did not occur during the tests and, hence, were not coded at all.

### 3.3. Introduction and Interaction Phases

#### Behavioural Reactions (Attitudes)

The three-way interaction between phase, human and treatment was not significant (χ^2^(1) = 0.41, *p* = 0.524). In a reduced model including all two-way interactions, only the human:treatment interaction was significant (χ^2^(1) = 4.59, *p* = 0.032; phase:human: χ^2^(1) = 0.23, *p* = 0.634; phase:treatment: χ^2^(1) = 1.17, *p* = 0.279). The final model only included the interaction between human and treatment to evaluate the main effect of phase (full-null model comparison: χ^2^(4) = 35.16, *p* < 0.001; see Appendix A). Dogs seemed to exhibit a more positive (more friendly, less insecure/offensive) reaction when their caregiver was greeting the fake dogs compared to the other human–treatment combinations (human:treatment interaction: χ^2^(1) = 4.51, *p* = 0.034; see Figure 4B). Moreover, the dogs’ reactions were significantly more positive in the Interaction phase compared to the Introduction phase (χ^2^(1) = 26.95, *p* < 0.001; see Figure 4A) and younger dogs tended to show more negative attitudes than older ones (χ^2^(1) = 8.53, *p* = 0.003; see Appendix A). Their reactions did not vary significantly between female and male dogs (χ^2^(1) = 0.42, *p* = 0.519).

### 3.4. Reaction Phase

#### 3.4.1. First Approach of Fake Dog

The interaction between human and treatment was not significant (χ^2^(1) = 0.17, *p* = 0.680). The reduced model without the interaction term (full-null model comparison: χ^2^(2) = 8.18, *p* = 0.017) revealed that dogs approached the fake dog first more often when the stranger was the demonstrator compared to the caregiver (χ^2^(1) = 5.58, *p* = 0.018; see Figure 5D). In contrast, there was no significant effect of treatment (χ^2^(1) = 2.77, *p* = 0.096) or the control predictors (see Appendix A).

#### 3.4.2. Friendly Interaction with Fake Dog

The interaction between human and treatment was not significant (χ^2^(1) = 0.49, *p* = 0.482). The reduced model without the interaction term (full-null model comparison: χ^2^(2) = 11.27, *p* = 0.004) revealed that dogs interacted with the fake dog in a friendly way more often when the caregiver was the demonstrator compared to the stranger (χ^2^(1) = 4.30, *p* = 0.038; see Figure 5C). Additionally, dogs interacted more with the fake dog following the greeting treatment compared to the medical check treatment (χ^2^(1) = 7.55, *p* = 0.006). The control predictors had no significant effect on dogs’ interactions with the fake dog (see Appendix A).

#### 3.4.3. Blocking Behaviour

The interaction between human and treatment was not significant (χ^2^(1) = 0.07, *p* = 0.799). The reduced model without the interaction term (full-null model comparison: χ^2^(2) = 6.57, *p* = 0.038) revealed that dogs showed the blocking behaviour more often when the caregiver interacted with the fake dog compared to interactions by the stranger (χ^2^(1) = 6.11, *p* = 0.013; see Figure 5A). In contrast, there was no significant effect of treatment (χ^2^(1) = 0.57, *p* = 0.451) or the control predictors (see Appendix A).

#### 3.4.4. Sniffing of Fake Dog’s Anal Region

The interaction between human and treatment was not significant (χ^2^(1) = 0.11, *p* = 0.736). The reduced model without the interaction term (full-null model comparison: χ^2^(2) = 6.65, *p* = 0.036) revealed that dogs sniffed more often at the fake dog’s anal region when the caregiver was the demonstrator compared to the stranger (χ^2^(1) = 5.84, *p* = 0.016; see Figure 5B). Treatment (χ^2^(1) = 1.36, *p* = 0.244) or the control predictors had no significant effect on dogs’ interactions with the fake dog (see Appendix A).

#### 3.4.5. Friendly Interaction with Human

The interaction between human and treatment was not significant (χ^2^(1) = 0.03, *p* = 0.853). The reduced model without the interaction term (full-null model comparison: χ^2^(2) = 6.06, *p* = 0.048) revealed no significant effects of the test predictors (human: χ^2^(1) = 3.64, *p* = 0.056; treatment: χ^2^(1) = 2.63, *p* = 0.105; see Figure 5E) or control predictors (see Appendix A).

#### 3.4.6. Behavioural Attitudes

The interaction between human and treatment was not significant (χ^2^(1) = 1.29, *p* = 0.256) nor was a reduced model without the interaction (full-null comparison: χ^2^(2) = 3.83, *p* = 0.148).

We also examined whether dogs that showed the blocking response had different behavioural ratings from those that did not show blocking. Indeed, blocking dogs had significantly different behavioural ratings compared to non-blocking dogs (Fisher’s exact test: *p* = 0.004) with dogs rated as “offensive” showing the blocking behaviour more frequently and dogs rated as “neutral” showing the blocking response less frequently (see Figure 5F).

Following, Table 4 shows a comparison between the predictions and the study results, according to the different hypotheses and test condtions.

## 4. Discussion

Corresponding to former survey studies investigating jealousy in dogs [1,2], in our study 70% of the caregivers (N = 102) reported that their dogs had shown jealousy-like behaviours, especially when the caregiver petted, praised, trained, or played with another dog, or when the other dog approached the caregiver or was close to him/her. The dog behaviours interpreted as jealousy were, for example, squeezing in-between the human–dog interaction (blocking the caregiver from the other dog/social rival), trying to separate the caregiver from the other dog, reacting aggressively by growling, snarling, barking, launching and snapping at the other dog, and trying to regain the caregiver’s attention and affection. Similar to these reports and other behavioural studies investigating jealousy-like behaviours in dogs, we also found that the dogs responded in a rather variable way to a third party their human caregiver was interacting with [7,8,23].

In addition to confirming the high prevalence of jealousy-like behaviours reported by caregivers in pet dogs, we investigated in the current study whether the behavioural responses of the same dogs in a situation frequently linked (by caregivers) to the occurrence of jealousy-like behaviours could be best explained by jealousy-like emotions or other mechanisms. To this end, we compared the dogs’ reactions when observing their caregiver or a stranger interacting with a realistic-looking fake dog either in a positive (greeting) or neutral (vet check) manner. We hypothesized that the dogs, if driven by jealousy, would show negative behavioural reactions (e.g., blocking behaviour) towards the fake dog, and would do this more often during the caregiver–fake dog greeting interaction. Even if a number of our findings did not support this “jealousy hypothesis” (see Table 4), we did find that, in total, more dogs showed blocking (getting physically between the human and the third party) than friendly behaviours towards the fake dog and that the dogs tried to block the caregiver’s interactions with the fake dog more often than those of the stranger. However, we found no evidence that blocking would have been more frequent in the case of positive rather than neutral interactions with the fake dog. Moreover, our results also raise the question whether blocking should (always) be seen as a behaviour indicative of jealousy. Alternatively, blocking may just come about because the dogs tries to position themselves in front of the human partners to be able to better interact with them (“direct response hypothesis”) or to better match the interaction the human is having with the frontal part of the fake dog (“synchronization hypothesis”). On the one hand, most dogs that approached the fake dog in an offensive manner also showed blocking, which lends support to the jealousy interpretation of this behaviour. On the other hand, approximately half of the dogs that approached the fake dog in a friendly or insecure manner also showed blocking, and only dogs that reacted in a neutral way showed no blocking. These findings suggest that blocking may often occur when the dogs are positively or negatively aroused. This is in line with the results of another study [23] which found that the dogs displayed no aggressive behaviours towards the furry and plastic fake dogs they presented to the subjects. In addition, in their second behavioural jealousy study [22] conducted with two (real) dogs of the same household, only 21% of the tested dogs (N = 42) tried to block the interaction of the caregiver and the companion dog.

In contrast to our predictions, that blocking should mainly occur in the caregiver greeting condition and be associated with negative behavioural attitudes, the prevalence of this behaviour was not confined to positive interactions between caregiver and fake dog or negative behavioural attitudes of the dogs. The blocking prevalence of about 50% in our two experimental groups was rather low compared to the frequency of reported jealousy-like behaviours within the same dog sample. A possible explanation of this finding can be two-fold: (a) the experimental setup of our study using a fake dog could have been too artificial and not sufficiently jealousy-evoking [22,23], and/or (b) it might be that the caregivers’ reports of their dogs’ supposedly jealous behaviours were predominantly based on anthropomorphising their dogs’ behaviour. Consequently, our findings highlight the need for behavioural experiments in this area of research. This could also help with improving the caregivers’ perceptions and evaluations of their dogs’ behaviours.

Nevertheless, using a fake dog as third party is often criticized since it is difficult to identify whether dogs perceive it as a real dog, and hence, as a potential social rival, or even as a threat to the dogs’ caregiver relationship. However, 34% of our tested dogs responded in an agonistic way (i.e., offensive or insecure-offensive attitude—behavioural category) to the entering fake dog, consistent with the notion that these dogs initially perceived the fake dog as a social threat. Additionally, when being released, the dogs sniffed significantly more at the fake dog’s anal region when the caregiver interacted with the fake dog. Interestingly, most of the dogs even repeatedly sniffed the anal region of the fake dog, and 22% of them showed further friendly social interaction with it. Another study using various fake dogs [23] also found that dogs socially investigated the fake dogs by sniffing the anal region and the muzzle (83% in study 1: with furry fake dog, 78% in study 2: with plastic dog). Moreover, very recently, another study [7] also showed that 14 out of 15 subject dogs (93%) displayed conspecific-directed behaviours, i.e., sniffed the fake dogs’ (facial and) anal regions when being unleashed after the last test trial (of only observing the interactions), and thus, claimed that they perceived the fake dogs as real dogs [7]. This is in line with the fact that anogenital sniffing is a common (social) behaviour in dogs to identify the sex or identity of another dog, e.g., when approaching or greeting conspecifics, and hence, might indicate that dogs perceived the fake dog as a real dog or social rival, at least before finally getting closer or physically interacting with it [8,58,59,60]. Another study [61] points in a similar direction, where they found that 292 pit bulls seized from illegal dog fighting operations reacted similarly aggressive towards real dogs and a dog-like model, but not to the control stimuli. Nevertheless, it would be worthwhile to conduct control conditions with self-propelled objects that do not have the appearance of a dog to show that the dogs’ reactions to fake dogs are indeed indicative of their perception of a conspecific, social rival.

Importantly, in our study, we chose a between-subject design in order to confront each subject with the fake dog only once, and part of the behavioural data reported here was collected before the dogs could approach and closely inspect the fake dog. An advantage of the fake dog is that we could create a highly controlled, standardized experimental setup and could avoid ethical concerns that would have arisen from exposing a real dog partner to the subjects’ possibly aggressive or ambivalent behaviour in an emotionally charged and space restricted setting.

In all three phases of our experiment (Introduction, Interaction and Reaction phase), we observed dogs whose behaviour could be categorised as friendly, neutral, insecure, insecure-offensive and offensive. During the Introduction phase, we found that the dogs responded rather neutrally or negatively (insecure reaction) while they were leashed and observing the fake dog and the human entering. Note that the direct, partly frontal approach and lack of communication of the fake dog, the intermittent noise by the wheels and the rather unnatural behaviour of the human partner (caregiver or stranger) may also have influenced the dogs’ initial reactions. Hence, we cannot fully disentangle whether they perceived the fake dog as a social threat or a novel object at first glance.

In the Interaction phase, when the human member of the triad was interacting with the fake dog but the subjects were still restrained from joining the interaction, we detected the least variation across dogs. Interestingly, seeing the humans interact with the fake dog evoked a clearly positive shift in the dogs’ behavioural reactions: almost all of them switched from neutral, insecure or offensive to neutral or friendly behaviours during the Interaction phase, as compared to the Introduction phase before. These positive reactions were particularly frequent when the dogs saw their caregiver greeting the fake dog, and, importantly, continued when the dogs were released to interact with the dyad. Even if many dogs might have perceived the first two phases of our experiment as a novel and ambiguous situation that was finally resolved by the human’s interaction with the fake dog, some of our findings suggest that the dogs reconsidered the situation as soon as they were released. One result pointing in this direction is that 20% of the dogs switched again to an insecure behavioural reaction when finally having the chance to freely interact with the human–fake dog dyad.

In the Reaction phase, we observed that more dogs acted friendly with the fake dog during the positive caregiver–fake dog interaction which supports the “behavioural synchronization hypothesis” (see Table 4). Furthermore, we found that the dogs exhibited a more positive behavioural reaction (more friendly and less insecure-offensive attitude) when their caregiver was greeting the fake dog as compared to the caregiver vet check group, whereas we observed no such difference between the two stranger groups. This human–treatment interaction again supports the behavioural synchronization hypothesis and is in line with the finding of another study [23] that the dogs interacted significantly more with different objects when their caregiver manipulated them, in comparison to a stranger. Albeit, based on former social referencing studies in dogs [45,46], we expected that this treatment effect would occur only in the caregiver groups. If behavioural synchronization explained the dogs’ behaviours in the Reaction phase, the behavioural reactions of the dogs in the first two phases of the experiment might have likely been driven by anxiety and uncertainty evoked by the rather surprising, unnatural entrance of the fake dog till the human started to interact with the fake dog, which seemed to have resolved this situation. These findings are in conflict with the “jealousy hypothesis” which predicted that the human–fake dog interaction would evoke a negative reaction towards the fake dog, especially when seeing the caregiver interacting in a positive way. Our results are consistent with another behavioural dog study investigating jealousy by using two different kinds of fake dogs and objects which also could not find clear evidence for jealousy-like behaviours in the tested dogs. In this study, the dogs were similarly paying attention to and interacting with their caregiver and the stranger who both handled the various objects [23].

We found little evidence that the dogs in the Reaction phase were motivated to adjust their behaviour directly to the one of their caregiver. Most dogs approached the fake dog first, instead of the caregiver, upon being released. Although 36% of the dogs did interact with the human in a friendly manner, 22% also did so with the fake dog. Neither the relationship with the caregiver nor their behaviour affected how many of the dogs engaged in such interactions with their human partner. Furthermore, none of the response variables showed a treatment effect specific to the stranger groups, which would have been a prediction of the “direct response hypothesis” (see Table 4).

In addition, we observed that younger dogs reacted more negatively (insecure, insecure-offensive, offensive) towards the fake dog (in Introduction and Interaction phase; see Appendix A) compared to older ones (more neutral, friendly). This matches with other findings [62] which demonstrated that juvenile dogs approached different test partners (a remote controlled car, AIBO robot, AIBO robot with a puppy-scented furry cover, 2-month-old puppy) earlier, and growled and/or barked more than adult dogs, especially towards the puppy-scented, furry and dog-like robot. This suggests that young dogs may be more novelty seeking, but also more insecure at the beginning of our test trials compared to older dogs. This is supported by a recent study investigating dogs’ personality changes over their lifetime that reported that novelty seeking in the tested dogs started to decrease in middle age [63]. The AIBO robot study [62] also reported that juveniles sniffed and oriented longer to the test partners than the adults. Another dog study on age effects stated the highest scores of explorative behaviours in young (2–4 years old) beagles in six diverse spontaneous behavioural tests [64]. For example, they more frequently sniffed at and interacted with the model dog (life-sized sandcast golden retriever model), a wall-mounted dog cardboard, and even reacted more to their own mirror reflection compared to aged dogs (9–15 years old).

## 5. Conclusions

To conclude, we found little evidence that the behavioural reactions of the tested dogs were driven and motivated by jealousy in this study. Although our results show that the dogs directed their behaviour to the fake dog in this triadic social interaction (instead of trying only to interact with the human), most of our results rather support the behavioural synchronization than the jealousy or direct response hypothesis. The dogs’ behaviours both during the observational phases (Introduction and Interaction phase) and when the dogs were free to interact with the dyad (Reaction phase) support this interpretation. Although alternative explanations are possible, these findings do not support the caregiver reports concerning jealousy. Our results suggest that the dogs’ interactions with other dogs and humans may be motivated by diverse emotions, and, importantly, might be more easily influenced and turned in a positive direction by the caregivers than they themselves may recognize. We cannot exclude, however, that in this sample of highly socialized dogs, jealousy-like behaviours might be less prevalent also due to the dogs’ training histories. Nevertheless, these findings suggest that it would be important to establish comprehensive knowledge about (individual) dogs’ emotional and behavioural expressions to prevent future misunderstandings and potential wrong interpretations or reactions towards the dogs themselves, while also ensuring the dogs’ wellbeing. This task of caregivers and researchers is not easy, however, as relevant behaviours may occur in rather subtle and short forms and are, therefore, difficult to assess. In this study, possibly because the dogs could interact with a fake dog, this was also obvious in the low interobserver reliability of coding both friendly and agonistic behaviours towards the fake dog. Hence, more research, in particular assessing additional non-behavioural measures such as physiological parameters (i.e., heart rate variability, breathing rates, skin temperature, oxytocin, cortisol levels, etc.), is needed to elucidate this interesting phenomenon in pet dogs. Additionally, future studies would benefit from more naturalistic study situations which might have an even greater potential for evoking jealousy-like behaviours.

## Figures and Tables

**Figure 1 animals-12-01574-f001:**
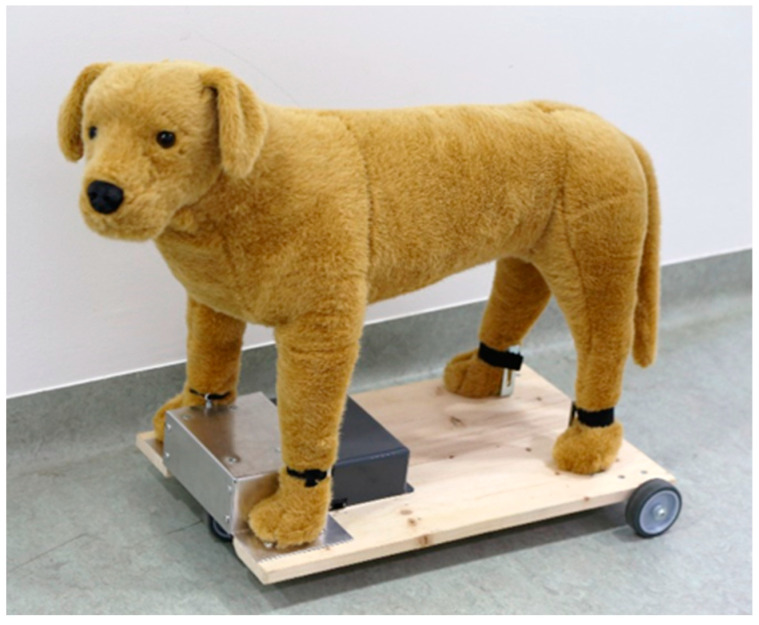
Fake dog mounted on board with wheels that could be moved via remote control.

**Figure 2 animals-12-01574-f002:**
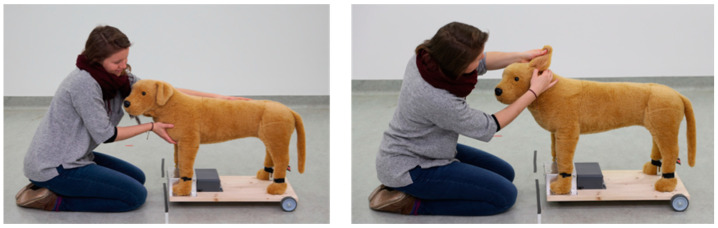
Human–fake dog interaction. Human (caregiver or experimenter) greeting (**left**) and vet check (**right**) performance with fake dog.

**Figure 3 animals-12-01574-f003:**
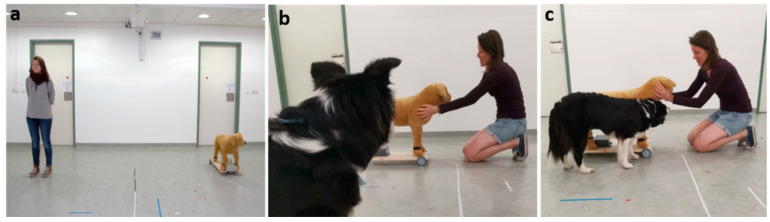
Different phases of each test trial (except for Exploration phase). Introduction phase (**a**), Interaction phase (**b**) and Reaction phase (**c**).

**Figure 4 animals-12-01574-f004:**
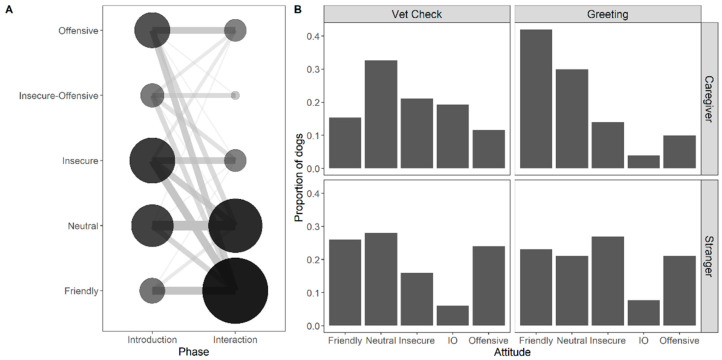
(**A**). Behavioural rating across the Introduction and Interaction phases. The size of the dots and the width of the lines are proportional to the number of represented dogs. (**B**). Bar plot showing the proportion of dogs in the different human–treatment conditions according to their behavioural ratings.

**Figure 5 animals-12-01574-f005:**
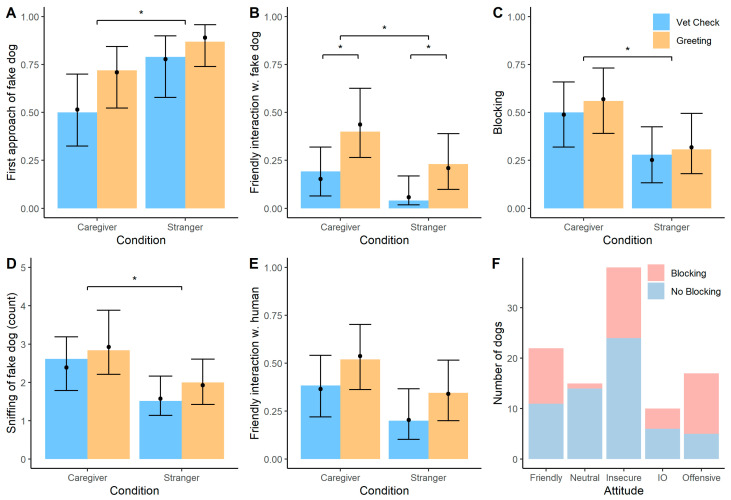
Bar plots showing different response variables in the reaction phase across the treatment and human conditions and the behavioural ratings. (**A**). Proportion of dogs that showed blocking behaviour. (**B**). Mean count of sniffing instances of the anal region of the fake dog. (**C**). Proportion of dogs that interacted in a friendly manner with the fake dog. (**D**). Proportion of dogs that first approached the fake dog (and not the human demonstrator). (**E**). Proportion of dogs that interacted with the human demonstrator in a friendly manner. (**F**). Number of dogs according to their attitude ratings in the reaction phase and divided into dogs that showed a blocking behaviour or not. The error bars represent 95% confidence intervals, the black dots show the model predictions, * GLM, *p* < 0.05.

**Table 1 animals-12-01574-t001:** Between-subject design of the experiment and number of dogs tested in the four test groups/conditions.

		Nature of Interaction	
		*Greeting*	*Vet Check*
**Human interacting with fake dog**	*Caregiver*	CG (N = 25)	CV (N = 26)
	*Stranger*	SG (N = 26)	SV (N = 25)

CP/SP—caregiver/stranger petting; CV/SV—caregiver/stranger vet check.

**Table 2 animals-12-01574-t002:** Coded dog behaviours in the respective test phases.

Test Phase	Coded Dog Behaviour or Attitude	Definition
Reaction phase	First approach to the fake dog (binary variable)	The first attempt of approaching the fake dog within 10 cm.
	Friendly interactions with caregiver/stranger (binary variable)	Dog interacted with the caregiver/stranger with relaxed body, ears and tail (often wagging their tail) either by leaning towards the person, licking her or initiating play by a play bow (stretching the body and front legs on the floor, tail wagging, could have been combined with barking). Note that small dogs sometimes used the human as a “ramp” to reach fake dog—this was not coded as interaction with human.
	Friendly interactions with fake dog (binary variable)	Dog interacted with the fake dog with relaxed body, ears and tail (often wagging their tail) either by leaning towards the fake dog, licking it or grooming it, or initiating play with a play bow (stretching the body and front legs on the floor, tail wagging, could have been combined with barking).
	Blocking (binary variable)	Any part of the dog was positioned in-between the fake dog and the caregiver/stranger.
	Offensive manipulation (binary variable)	Dog manipulated the fake dog with its mouth (biting or grabbing), paws (put paw on it) or body (leaning towards it) with stiff body, head held high, tail position above back level. This could be combined with growling.
	Dominant behaviours (binary variable)	Stand tall: dog straightened up to full height with a rigid posture and tail, may include raised hackles, ears erect and tail perpendicular or above the back. Dominant approach: to approach fake dog within one meter for at least 5 s with the tail perpendicular or above the plane of the back and the ears erect and pointed forward. Paw on: to place one or both front paws on the fake dog’s back. Head on: Dog approached fake dog with a rigid posture and puts its head on its shoulder/back. Most of the time formation looked like a capital “T”. The tail was usually held up or perpendicular to the body.
	Sniffing anal region of fake dog (count variable)	Dog’s nose was a few centimetres away from the anal region of the fake dog (a few centimetres around the tail), either with acoustically hearable sniffing sounds or with small, fast head movements to initiate sniffing.

**Table 3 animals-12-01574-t003:** Coded dogs’ overall behavioural reaction to fake dog in the respective test phases.

Test Phase	Dogs’ Overall Behaviour	Definition
All phases	Friendly	The dog’s body posture was relaxed, no piloerection (erected hair), ears were loose or slightly pointed forward, tail was around the level of the dog’s back and was wagging or hanging in a relaxed manner. Could have been accompanied by barking, jumping and running.
	Neutral	Dog showed no interest towards the fake dog or caregiver/stranger, did not look intensively at it (stared for longer than 3 s with stiff body) or went in the direction of it. Body language was calm and body parts/muscles were loose, tail was low and relaxed. The dog was standing, sitting, lying and/or sleeping, gaze direction was often changing between fake dog and caregiver/stranger.
	Insecure	Dog stepped, moved or ducked away from the fake dog (or made attempts to do so when being on leash). The body posture was slightly crouched, ears pointed backwards, tail was lower than the level of the dog’s back or even between the legs. Occasionally it was combined with avoiding to look in the direction of the fake dog, with showing attention/support seeking from caregiver/stranger, displaying stress signals and/or barking. If the dog approached the fake dog, it moved towards the fake dog in an indirect route, following a curve with slightly crouched body.
	Insecure-offensive	Dog stared (longer than 3 s) at the fake dog, moved towards or away from it (or made attempts to do so when being on leash), body slightly crouched, tail was lower than the level of the dog’s back. It was occasionally combined with ears pointing backwards, growling or barking.
	Offensive	Dog actively moved towards the fake dog or leaned into the leash/harness or stared at it (longer than 3 s) while its body language was stiff, ears erected and tail was above the back, erected or sometimes slowly wagging. Could be combined with growling or barking, or rushing up to the fake dog, snapping or biting.

Notes: The Introduction, Interaction and Reaction phases were coded throughout all trials; overall, the dog’s behavioural reaction towards the fake dog throughout the Introduction and Interaction phases and their initial reaction off-leash during the Reaction phase were coded; if a dog exhibited multiple behaviours that fell into different categories, then emotional expressions calling for an offensive categorization were weighted higher than insecure-offensive expressions; if insecure-offensive and insecure behaviours occurred, then category of insecure-offensive was coded; insecure was rated over friendly, and friendly over neutral (i.e., neutral < friendly < insecure < insecure-offensive < offensive).

**Table 4 animals-12-01574-t004:** Overview of the hypotheses, predictions, dependent variables and the results supporting each hypothesis.

		*Hypotheses*		
		Jealousy	Behavioural Synchronization	Direct Response to Human
** *Results* **	Predicted dogs’ reactions and group effects	Negative reaction (i.e., blocking behaviour) towards fake dog; caregiver > stranger, greeting > vet check	Positive reaction (i.e., friendly approach) towards fake dog; partner–treatment interaction: petting > vet check, only for caregiver	Positive reaction (i.e., friendly approach) towards human; partner–treatment interaction: petting > vet check, only for stranger
** *Predictions* **	Study results supporting respective hypotheses	More blocking in caregiver greeting group and in friendly, insecure and offensive dogs; more dogs approached caregiver first; increased sniffing of fake dog	Positive response to human–fake dog interaction; more dogs approached fake dog first; more friendly interactions with the fake dog and the human, especially in the positive caregiver group	Positive response to human–fake dog interaction; less friendly interactions with the human, especially in the neutral groups

## Data Availability

Data sets and statistical analyses supporting the reported results can be found here: https://github.com/cvoelter/fake_dog_interactions (accessed on 27 May 2022).

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
