# Peer review of "Pet dogs’ Behavioural Reaction to Their Caregiver’s Interactions with a Third Party: Join in or Interrupt?"

_animals, 2022, doi:10.3390/ani12121574_

Round 1

Reviewer 1 Report

This manuscript addresses whether family dogs show jealousy-like behaviour  in situations in which the owner or a stranger interact with a fake dog positively or in a neutral way.

Overall find that it is an interesting work, adding to the literature on this topic. I particularly find it interesting and novel that dogs seem to also synchronize / match their behaviour to that of the human, in this situation.

The main concern is I have is about the use of a fake dog and particularly, the fact that this object does not move, thus cannot react to the human greeting / vet-check. This, obviously, may affect the results. However, the authors justify this choice (which has also been made by other authors in the past) and also acknowledge the limitations appropriately in the text.

Overall, I think this work is worth of publication, after minor revision.

My detailed comments are listed below: 

The intro would benefit of a clear description of what are the behavioural components of jealousy-like behaviour. This way the reader woud have a clearer view about which of them were expected and  found in this study.

Lines 120-126: Another reason for the “jealousy bias” may be that that studies conducted so far did not succeed in experimentally reproduce natural situations in which the jealousy-like behaviour is elicited (e.g., mostly they used fake dogs...)

Line 137: I think it is difficult to test “if they experienced jealousy”. Maybe the authors refer to whether the dogs performed behaviours that, in humans, would be consistent with this feeling? (here, if there was a behavioural description of jealousy-like behaviour, this could be used).

Line 140-142 would better fit after the predictions.

Lines 260-261: How were the owners talking to the fake dog? (What tone of voice, what emotional cues they expressed?)

How different was this in the vet check and in the greeting conditions?

It may be difficult for owners to pretend to talk to a fake dog and use neutral or emotionally loaded cues / tone of voice etc.

Was there really a clear, precievable difference between the two conditions, in this respect?

Line 301-304: This seems rather far from any real-life situation.

Could this have affected how the dogs perceived the fake dog? (fake VS real)?

Line 343: inter-rater agreement for “friendly interaction with fake dog” seems to me quite low. This should be considered carefully as a limitaion of the study.

Lines 357-358: This is not very clear. Does this (>) indicate temporal sequence or one category being of longer duration than others? Or…?

Lines 402-404: This text would better fit in the methods

Lines 534-535: Could not it also be that an interaction like the vet-check between owner and stranger dog is not a situation that the subjects observed very frequently and, therefore, they still reacted in a way that is similar to (more frequent) positive interactions between owners and stranger dogs? Probably most family dogs do not often observe other dogs being vet-checked.

Lines 541-542: It may also be that jealousy-like behaviours also manifest together with other (more positive) behaviors, in a flexible manner, even during the same interaction.

Lines 566-567: could not it be that the dogs sniffed at the anal region of the fake dog right to assess whether it was a dog? (e.g., a dog that is completely immobilized whe greeted may appear quite unusual...).

22% of the dogs tried to socially interact with the fake dog: this is not a very high percentage! Almost 80% of the dogs did not do it…

Lines 612-614: This could also be due to the artificial / strange situation in wich the fake dog does not move, is stiff, has no dog odor etc.

Conclusions: Just something the authors may want to consider: Do synchronization and jealousy-like behaviour necessarily exclude each other?

Maybe the interactions are more nuanced and flexible than we think, even within the same individual. 

Reviewer 2 Report

Rev 1: Pet dogs’ behavioural reaction to their caregiver’s interactions with a third party: join in or interrupt?

The aim of this study is investigating dogs’ behavioural responses during different interaction (greetings vs medical examination) performed by caregivers or stranger and a fake dog.

A great job was done by the authors and I very appreciated this study. The exploration of animals’ emotions such as jealousy is an important issue in animal science and this study has proven to be adequate and exhaustive in adding knowledge about this topic. Moreover, the used approach was not based on the information given by the owners about jealousy but tried to bypass this by implementing different experimental tests.

Please, considering the following comments to improve the manuscript.

I suggest to revise the title focusing on jealousy and adding emotion in key words

About the aims, their description is very discursive and dispersive, I suggest to simplify this part in order to facilitate the understanding of the aims of the study (e.g. enter a numbering of aims).

LL 131-135 this part should be moved in materials and methods.

LL 150-191 This part seems too long and it seems more suited to the discussion.

LL 203-207 It could be useful a table on the different tests.

LL 210 experimental set up. It is not totally clear the different role of experimenters. Caregivers mean owners of dogs? Is E1 the stranger? (see LL 257). How many people were in the experimental room? Also, in this case a table summarizing the role of people involved in the experiments could be useful.

LL 341-348 this part should be moved in the results  

LL 393 It could be better to proceed following the sequencing of the different steps of the study. So, the part on questionnaire should be move at the beginning of material and methods part. It could be also important adding more information about the developed questionnaire (type of questions, structured, semi-structured questionnaire etc.), possible validation or adaptation from previous questionnaire, and on the statistical analysis performed eventually.

Statistical analysis: what about the distribution of the data?

Figure 4: the graph A is partially cut (dots at the extreme are cut)

LL 536 it could be interesting adding possible motivation of blocking behaviours different than jealousy.

LL 571 -572 add bracket

LL 631-632 please add reference if available  

Sorry but I did not find the supplementary materials (only table on dogs’ demographics).

Reviewer 3 Report

I thoroughly enjoyed reading the manuscript and found it very interesting. This is a very interesting paper, and I look forward to seeing it published. Most of my comments are only minor.
This appears to be a major development in the
attachment-related and potentially jealousy like behaviours in non-human animal field, and one which I was privileged to read. I praise the authors on an excellent study, and a good manuscript, and offer my thanks for this study, and offer my best wishes for the future.

You have a really nice abstract
Introduction
The introduction is nice, clear and well written. It has a nice flow and clearly states the aims of the study. Perhaps a little bit of extra detail on the inter-rater reliability of the test assessment
 maybe useful (or in conclusion?), but I do not feel strongly about that, and I will leave that up to your judgement.

Again, generally well written, but a few minor points:

-          The first and the second columns of the table 1 should be narrowed to allow you to reduce the rows of the third column

-          Line 385 I would delete the text in parenthesis

-          Line 418 These percentages are duplicated in the text, delete here

-          Line 421 Probably these behaviours are not exclusive, but I would expect the sum of the percentages to be 100. Unfortunately I was not able to open table S1, where it is probably better explained.

-          Lines 543, 626, 650, 657: please replace year with reference number!
